

# Clinical outcome of endonasal endoscopic prelacrimal approach in managing different maxillary pathologies

Yu Hsuan Lin[1,3,4] and Wei-Chih Chen[2]

[1] Department of Otolaryngology, Head and Neck Surgery, Kaohsiung Veterans General Hospital, Kaohsiung, Taiwan
[2] Department of Otolaryngology, Kaohsiung Chang Gung Memorial Hospital and Chang Gung University College of Medicine, Kaohsiung, Taiwan
[3] Department of Otolaryngology, Chung Shan Medical University Hospital and School of Medicine, Taichung, Taiwan
[4] Department of Otolaryngology, Head and Neck surgery, National Defense Medical Center, Taipei, Taiwan

Corresponding author
Wei-Chih Chen,
jarva@adm.cgmh.org.tw

## ABSTRACT

**Background:** The aim of the study was to evaluate the treatment outcomes of endoscopic prelacrimal recess approaches (EPLAs) in managing different sinus pathologies, analyzing associated adverse events and post-treatment quality-of-life.
**Methods:** We enrolled 21 consecutive patients (22 lesions) who received endoscopic sinus surgical procedures with EPLAs in two tertiary medical institutes between 2015 and 2018. Quality-of-life and self-rated symptom severity data were collected using the 22-item Sino-Nasal Outcomes Test (SNOT-22) and 10-point visual analog scales (VAS), respectively.
**Results:** A total of 21 patients (mean age (standard deviation) 51.7 (14.5) years; 16(76.2%) male) were followed up for 12.7 months. The most common symptoms were nasal discharge and nasal airway obstructions. Nine lesions (40.9%) were sinonasal papilloma's, seven lesions were other types of neoplasms (31.8%; five benign and two malignant), two were trauma-related (9.1%), and four inflammatory diseases (18.2%). Patients with non-papilloma lesions had higher presurgical SNOT-22 than those with papillomas (*P*-value = 0.021). After EPLAs, non-papilloma patients had significant improvements in SNOT-22 and VAS (*P*-values = 0.012 and 0.012, respectively), while those with papillomas had only marginally significant improvements in VAS (*P*-value = 0.061). The most common adverse events was temporary cheek/tooth numbness (*n* = 11), and patients with sinonasal papillomas were more likely to have post-treatment complications than those with other disease entities.
**Conclusions:** EPLAs were found to effectively manage various sinus diseases. Short-term life-quality improvements were promising. Future large-scale studies with longer follow-up periods are recommended.

## INTRODUCTION

Maxillary sinus is the largest of the paranasal sinuses (*Tomenzoli et al., 2004*). It is bordered posteriorly by the pterygopalatine fossa and infratemporal fossa, inferiorly by the alveolar process and superiorly by the orbital floor (*Tomenzoli et al., 2004*). Unilateral maxillary occupying lesions encompass a broad array of pathologies, predominantly sinusitis, followed by sinus cyst, and then benign neoplasms (mostly sinonsal papillomas) (*Coleman et al., 2005*). Because the maxillary sinus is the most frequent site of sinus pathologies, mere ostium opening by endoscopic approach may not adequately address all the problems (*Kennedy & Adappa, 2011*).

Most diseases of the maxillary sinus can be managed by endoscope, which is used mostly to perform standard middle meatal antrostomy (MMA) (*Kennedy & Adappa, 2011*). For benign lesions such as inverted papillomas, endoscopic medial maxillectomies are often performed for better operating field and management of the tumor attachment site (*Turri-Zanoni et al., 2017*; *Wormald et al., 2003*). While it is almost impossible to resolve every detail of the disease with conservative approach, more extended procedures are performed at increased risk of surgery-related morbidities (*Turri-Zanoni et al., 2017*; *Wormald et al., 2003*; *Lombardi et al., 2011*; *Bertazzoni et al., 2017*). *Nakamaru et al. (2010)* introduced the surgical creation of a corridor made by breaking a hole through the medial wall of prelacrimal recess (PLR), which is usually one of the most difficult positions to manipulate during standard MMA. This surgical corridor can provide an unobstructed view of almost any aspect of the maxillary inner linings. This improved visualization of the operating field can reduce the risks associated with endoscopic procedures (*Wormald et al., 2003*; *Lombardi et al., 2011*; *Bertazzoni et al., 2017*).

*Zhou et al. (2013*, *2016)* have promulgated the use of the endoscopic PLR approach (EPLA) to manage a variety of pathologies of the maxillary sinus and deep areas of the skull base. The use of the EPLA has been reported to provide results comparable to those of conventional external procedures (*Lee et al., 2019*). However, its application has mostly been focused on the removal of neoplasms and no studies have been performed to assess post-treatment life-quality following EPLA (*Lee et al., 2019*; *Zhou et al., 2018*; *Yu et al., 2018*; *Lin, Lin & Yeh, 2018*). Therefore, in this study, we retrospectively reviewed the clinical applications of endoscopic PLR approach in the surgical treatment of different maxillary pathologies. Patients were followed up to assess functional treatment outcomes (patient-reported overall health and symptom severity) as well as adverse events and need for revision surgery.

## MATERIALS AND METHODS

### Study design and patient eligibility

We identified 21 adult patients receiving surgical procedures utilizing the endonasal endoscopic prelacrimcal recess approach (EPLA) from February 2015 to October 2018 in two tertiary care referral centers retrospectively. These patients may have also had additional endoscopic sinus surgeries or deviated septum corrections when deemed necessary by a surgeon based on pathology, extent of disease, or a patient's specific

complaints prior to surgery. All patients enrolled in this study were recommended EPLAs because their diseases had been considered too difficult to treat using standard MMA alone. The protocol of this study was approved by the institutional review board of Kaohsiung Chang Gung Memorial Hospital (IRB No: 201900695B0). Written consents of the participants are not needed because of the retrospective nature of our study design.

All patients received high-resolution sinus computed tomography scans (1.0-mm slices in the axial view) to assess degree of inflammation. Sinus disease was scored based on modified-Lund–Mackay scores (*Snidvongs et al., 2014*) of the diseased side and further categorized using the Krous staging system (*Krouse, 2000*) or American Joint Committee on Cancer (AJCC) TNM classifications, depending on whether they were sinonasal papillomas or malignancies. PLR space was measured as follows. Antero-posterior (AP) diameter was first defined as the distance from pyriform aperture to the nasolacrimal duct (NLD) by identifying the inferior-most aspect of NLD on coronal view and then transposing that to axial view. The height of medial PLR was measured starting at the level of the superior-most aspect of the NLD to the nasal floor, as described in *Kashlan & Craig (2018)*. Other demographic data including age, gender, initial symptoms and their durations, image findings and known surgical history of each patient were also collected from each patient prior to surgery. After surgery, patients received a series of follow-up examinations in 3–6 visits spaced 1–4 weeks apart, depending on endoscopic findings and specific needs of the patient. Daily saline irrigation and intranasal corticosteroid spray were performed for post-treatment cares. Short-term oral medications were prescribed only for symptomatic patients. Revision surgery was performed if disease recurred.

## Outcome assessment

We assessed interval changes in patient-reported outcomes between initial baseline evaluations and the last follow-up date. Functional outcome measurements included self-rated visual analog scale for current overall health (zero being worst imaginable and 10 as best imaginable) and self-rated 22-item Sino-Nasal Outcome Test (SNOT-22) to assess sinonasal symptom severity (*DeConde et al., 2015*). Other outcomes were incidence and characterization of treatment-associated adverse events and need for revision surgery. Follow-up time was defined as the time between the first visit with an otolaryngologist for diagnosis to the last documented follow-up date.

## Surgical technique

All surgical procedures were performed with patients under general anesthesia. EPLA was performed before the other procedures for patients scheduled to receive endoscopic PLR approach for maxillary sinus lesions. To perform these procedures using the PLR approach, we first administered local anesthesia to the axilla of inferior turbinate using a 4 mm, 18 cm length zero degree rod-lens endoscope (Karl Storz Endoscopy, Tuttlingen, Germany). A 15-blade was used to create a curvilinear incision from the lateral nasal cavity to the nasal floor, pathway crossing just anteriorly to the head of inferior turbinate. The inferior turbinate-nasolacrimal duct (IT-NLD) flap was then elevated subperiosteally

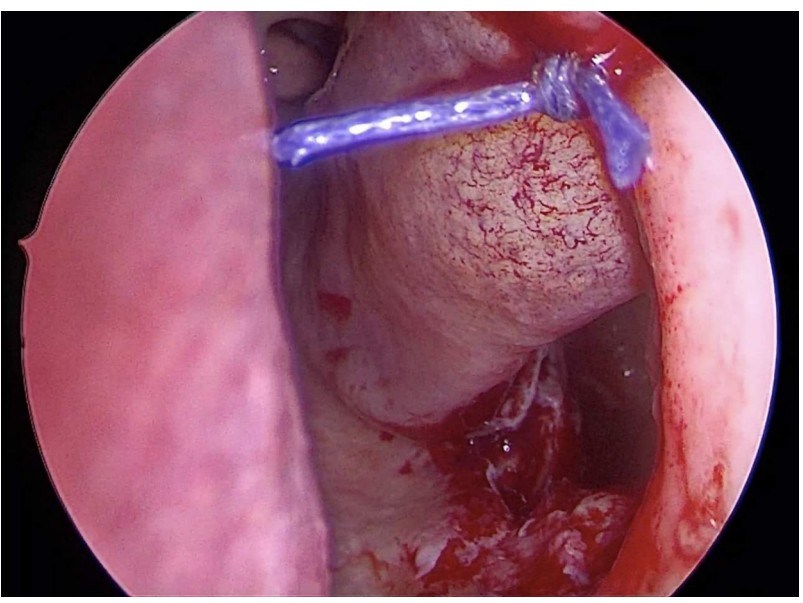

**Figure 1 Inferior antrostomy is created after pre-lacrimal approach combined with medial maxillectomy.** Inferior antrostomy was created to facilitate post-operative follow-up in inverted papilloma patients with risk of recurrence.

by a suction elevator pulling it toward the mucosal portion of the naso-lacrimal duct and placed medially. Using a 4 mm chisel to remove the head of inferior concha bone and part of frontal process of maxillary bone, we penetrated the antero-medial aspect of maxillary sinus. The entrance was further enlarged using a Kerrison Rongeurs and/or drilling burr. When operating space was not easily accessible and a wider working area was needed, we drilled out part of the anterior maxillary wall to make way extent our surgical fulcrum laterally. All the diseased mucosa as well as neoplasms were removed. The underlying hyperplastic bones were drilled out and cauterized electrically in patients with sinonasal papillomas (*Healy et al., 2016*). Additionally, the medial bony compartment was taken out altogether when the lesions were closely attached to the natural ostium and/or medial aspect of maxillary sinus. For inflammatory lesions, much of the underlying sinus mucosa was preserved as possible while the nidus were meticulously trimmed. The IT-NLD flap was re-draped onto its primary position suturing the incision site with 4-O vicryl. In some patients, an inferior meatotomy was created to facilitate future observation during following visits (Fig. 1).

## Statistical analyses

Categorical variables were expressed as frequency and percentage. Descriptive parametric data were expressed using mean, standard deviation (SD), and range. We used median and interquartile range (IQR) to express subjective metrics (VAS and SNOT-22 included). Their interval changes between different groups were tested using Wilcoxon signed rank test. The differences between two groups were tested using Mann–Whitney $U$ test. $P$-value < 0.05 was considered significant. All statistical operations were performed using IBM SPSS 22.0 statistical software (IBM Corp. Released

**Table 1 Demographic characteristic of 21 patients with underwent endonasal endoscopic prelacrimal recess approach.**

| Variables | Sinonasal papilloma $n = 9$ | Non papilloma $n = 12$ |
|---|---|---|
| Age, mean (SD) | 57.0 ± 15.4 | 47.8 ± 13.7 |
| Gender, $n$ (%) | | |
| Male | 7 (77.8) | 9 (75) |
| Female | 2 (22.2) | 3 (25) |
| Initial symptoms, $n$ (%) | | |
| Nasal airway obstruction | 4 (44.4) | 3 (25) |
| Nasal discharge/discolored post nasal drips | 2 (22.2) | 6 (50) |
| Facial & head pain/pressures | 0 | 5 (41.7) |
| Hyposmia/anosmia | 0 | 1 (8.3) |
| Blood tinged sputum/rhinorrhea | 5 (55.6) | 1 (8.3) |
| Ophthalmologic manifestations | 0 | 2 (16.6) |
| Asymptomatic | 1 (11.1) | 0 |
| CT Lund-Mackay scores, mean (SD) | | |
| Maxillary sinus (modified) | 3.0 ± 0.7 | 2.9 ± 1.2 |
| Total | 5.7 ± 6.8 | 4.6 ± 5.0 |
| Measured nasolacrimal complex parameters, mean (SD) | | |
| Antero-posterior dimension (mm) | 5.2 ± 1.3 | 5.6 ± 1.5 |
| Height (mm) | 26.4 ± 3.7 | 27.3 ± 5.0 |
| Revision procedures, $n$ (%) | 4 (44.4) | 3 (25) |
| Complications, $n$ (%) | | |
| Transient dental/cheek numbness | 9 (100) | 2 (16.7) |
| Epistaxis | 1 (11.1) | 0 |
| Delayed wound healing | 1 (11.1) | 0 |

**Note:**
CT, computed tomography; SD, standard deviation.

2013. IBM SPSS Statistics for Windows, Version 22.0. Armonk, NY: IBM Corp., Chicago, IL, USA).

# RESULTS

## Patients characters

A total of 21 patients (16 males and five females) with 22 lesion sites received surgical procedures utilizing the endonasal endoscopic prelacrimcal recess approach (EPLA) at our two hospitals. The mean (standard deviation (SD)) age at diagnosis was 51.7 (14.5) years (range 17–81 years) and mean follow-up after surgery was 12.7 months (range 1.4–41.5 months). Fourteen patients underwent these procedures as primary surgeries; the other seven received these procedures as revisions (Table 1). The other procedures included septoplasty ($n = 2$), endoscopic sinus surgery ($n = 14$), and inferior meatotomies ($n = 3$). Inferior meatotomies were created for facilitating postoperative investigation in two patients with diffuse papilloma and in one patient with trauma-related

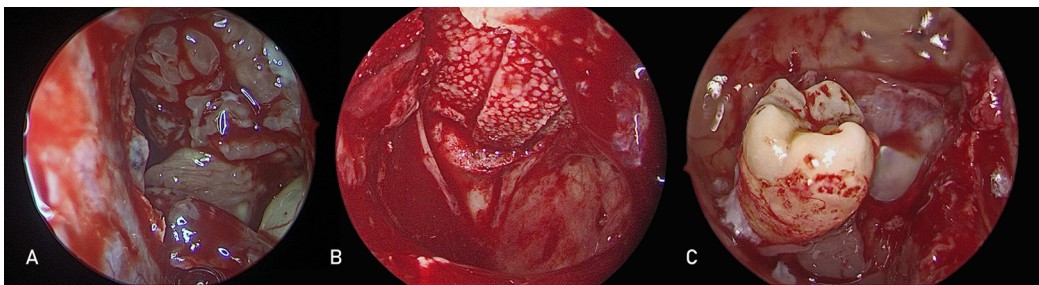

**Figure 2 Pre-lacrimal approach is used to treat different maxillary sinus pathologies.** Endoscopic pre-lacrimal approach for (A) left maxillary sinus inverted papilloma (B) left orbital floor defect repair with medpor (C) right maxillary sinus ectopic tooth removal.

**Table 2 Characteristic features of patients with sinonasal papilloma.**

| Patient number | Gender | Age | Pretreatment VAS | Pretreatment SNOT-22 | Disease extent within maxillary sinus (disease side) | Follow-up time, months | Previous operation |
|---|---|---|---|---|---|---|---|
| 1 | M | 59 | 8 | 17 | M, P (L) | 6 | N |
| 2 | M | 56 | 1 | 2 | M, P (L) | 17 | N |
| 3 | M | 30 | N/A | N/A | M, A, L (R) | 37 | Y |
| 4 | F | 49 | 3 | 27 | P, L (L) | 7 | N |
| 5 | M | 44 | 5 | 18 | I, M (L) | 7 | N |
| 6 | M | 71 | 0 | 2 | A, I (L) | 7 | N |
| 7 | F | 54 | 10 | 85 | A (B) | 42 | N |
| 8 | M | 81 | 4 | 17 | ALL (L) | 41 | Y |
| 9 | M | 69 | 6 | 20 | A, L, P (R) | 20 | Y |

Note:
VAS, visual analog scales; SNOT-22, 22-item Sino-Nasal Outcomes Test; M, medial; P, posterior; L, lateral; A, anterior; I, inferior; N/A, not available; N, no; Y, yes.

maxillary mucocele. Five (71.4%) of these seven revision patients received at least two interventions (Table 1). As can be seen in Table 1, a summary of initial presentations, the most frequent symptoms were nasal obstruction and nasal discharge followed by blood streaks within nasal drainages/blood tinged sputum. The duration of symptoms ranged from 30 days to 72 months (mean (SD): 17.2 (24.2) months).

## Pathologies and imaging features

Patient characteristics and disease features are summarized in Table 1. Three of the patients (14.3%) had inflammatory sinonasal disorders (two primary chronic rhinosinusitis and one recurrent bilateral CRS with nasal polyps). Eleven patients had benign neoplastic lesions within the maxillary sinus. Of these 11 patients, nine patients (81.8%; eight unilateral, one bilateral) had sinonasal papillomas (nine inverted, one exophytic) (Fig. 2A), one had a sinonasal organized hematoma, and the other had a cavernous hemagioma. Three patients had odontogenic cysts, including two patients with comorbid maxillary ectopic teeth (Fig. 2B). Within the remaining four patients, two

**Table 3 Outcomes of patients treated with endoscopic prelacrimal recess approach.**

|  | Pre-treatment | Post-treatment | P-value |
|---|---|---|---|
|  | VAS, median (IQR) | VAS, median (IQR) |  |
| Papillomas | 4.5 (1.5–7.5) | 1.5 (0–3.5) | 0.061 |
| Non-papillomas | 8.8 (7.3–10.0) | 1.8 (0.3–3.2) | 0.012 |
| P-value | 0.050 | 0.867 |  |
|  | SNOT-22, median (IQR) | SNOT-22, median (IQR) |  |
| Papillomas | 17.5 (5.8–25.3) | 8.0 (3.3–11.0) | 0.176 |
| Non-papillomas | 37.5 (26.5–50.3) | 13.5 (10.8–28.8) | 0.012 |
| P-value | 0.021 | 0.121 |  |

Note:
    VAS, visual analog scales; SNOT-22, 22-item Sino-Nasal Outcomes Test; IQR, interquartile range.

patients presented trauma-related disorders (an orbital fracture requiring medial and inferior orbital repair as seen in Fig. 2C and a maxillary mucocele that developed decades after facial trauma) and the other two patients with individual malignancies (lacrimal duct adenoid cystic carcinoma and recurrent buccal cancer with involvement to the infratemporal fossa) both underwent pathology obtainment by EPLA. The extent of disease and locations of the papillomas within maxillary sinus can be found in Table 2.

Average AP diameter was 5.5 (1.4) mm, and mean PLR medial wall height 26.9 (4.2) mm (range 19–34.9 mm). AP diameter of three diseased sides is greater than 7 mm whereas the measurement of the remaining 19 sides ranged from 3 mm to 7 mm. We did not find difference of modified-Lund–Mackay scores between patients with sinonasal papillomas and chronic rhinosinusitis (3.0 (IQR 2.0–3.5) vs. 4.0 (IQR 4.0–4.0), $P = 0.064$). The mean (SD) values for pretreatment SNOT-22 and VAS are 32.6 (22.1) and 6.7 (3.4), respectively. Patients who did not have papillomas reported statistically higher VAS ratings and SNOT-22 scores than those with papillomas (Table 3).

## Outcomes and complication profiles

Within these 21 patients, five patients did no complete subjective outcome measurements (SNOT-22 and VAS). Finally, 16 patients were enrolled in functional outcomes analysis measured at the last clinical visit after primary EPLAs. There were no significant differences in post-treatment patient-reported outcomes between those with sinonasal papillomas and those with other pathologies (Table 3). Patients who did not have papillomas reported significantly improved VAS ratings and SNOT-22 scores (Table 3, both P-values < 0.05), while individuals with papillomas also reported improved VAS ratings and SNOT-22 scores, though without statistical significance (Table 3). There was no disease recurrence during the follow-up period.

A total of 11 patients had 13 adverse events. Papilloma patients had a significantly higher incidence of post-operative complications, compared to patients with other pathologies ($n = 11$ vs. $n = 2$). The most commonly seen complication was paresthesia ($n = 11$) of cheek and tooth, and all symptoms of this complication resolved within 6 months. Patients with AP diameter ranged from 3 mm to 7 mm tend to have higher

prevalence of surgery-associated neural sequels, compared with individuals with AP diameter greater than 7 mm (10/19, 52.6% vs. 1/3, 33%, respectively), though of no statistical significance ($P = 1.00$). One patient had delayed wound healing resulting from a prior submucosal turbinectomy allegedly caused by post-surgical scarring. Additionally, one patient developed epistaxis soon after the surgery. No nasal lacrimal ducts or negative esthetic issues (i.e., nasal alar collapse/soft tissues depressions) were found.

## DISCUSSION

This study found that EPLA-treated patients reported improved VAS ratings and SNOT-22 scores over time, with mean improvements of 4.2 and 17.6 points, respectively. These scores suggest significant improvement in quality of life after EPLAs. Although we encountered treatment-associated adverse events, most of them are paresthesia occurred in inverted papilloma cases. However, all adverse events were all manageable and the sensory deficits subsided within 6 months. These findings suggest that the endoscopic PLR approach can be used safely and effectively to treat a variety of maxillary sinus pathologies.

In the surgical treatment of diseases originating from different sites of the maxillary sinus, the endoscopic pre-lacrimal approach makes possible the management of almost any aspect of internal linings. Especially the anterior and inferior aspect (e.g., alveolar recess and PLR), which are locations that traditional surgical corridors can hardly access (*Nakamaru et al., 2010*; *Zhou et al., 2013*, *2016*; *Lee et al., 2019*; *Zhou et al., 2018*; *Yu et al., 2018*; *Lin, Lin & Yeh, 2018*). In select cases in which disease is located far laterally, surgeons can drill the pyriform aperture and/or part of anterior maxillary wall to extend surgical fulcrum (*Turri-Zanoni et al., 2017*; *Nakamaru et al., 2010*; *Zhou et al., 2013*).

One recent study has reported that operations employing EPLA to treat maxillary or retro-maxilla lesions can achieve outcomes comparable to those achieved by traditional open approaches performed by experienced hands (*Zhou et al., 2016*; *Lee et al., 2019*). However, before we try to implement this technique more widely, some of its disadvantages need to be understood. The first being that the inferior turbinate-nasal lacrimal duct mucosal flap is re-draped onto its original position, the anterior half of the maxillary sinus (especially the alveolar recess inferiorly and zygomatic recess laterally) is difficult to be evaluated even with the use of flexible endoscopy post-operatively (*Zhou et al., 2018*). Therefore, additional inferior meatotomy may be necessary for full-filled surveillance for patients with high risk of disease recurrence (*Zhou et al., 2018*). Further, post-surgical debridement and saline delivery are less convenient in comparison with standard endoscopic medial maxillectomy (*Turri-Zanoni et al., 2017*). Still another challenge is that EPLA must be very carefully performed when treating neoplasms involving or infiltrating the inferior turbinate, nasolacrimal complex or medial wall of the maxillary sinus (*Nakamaru et al., 2010*; *Zhou et al., 2013*). In addition, overzealous manipulation of mucosa within alveolar recess should be avoided to prevent inadvertent trauma to the minor branches of anterior superior alveolar nerve (branches from the infraorbital nerve running within the maxilla). Care must also be taken to avoid dental injury because first and second molar tooth are dehiscent into the maxillary sinus, which has been reported to occur in 2% of normal population (*Coleman et al., 2005*).

Initially EPLA for neoplastic lesions were designed to provide a corridor to allow the management of lesions within infratemporal and pterygopalatine fossa tumors (*Zhou et al., 2016*). Recently, its use has been extended to the treatment of maxillary neoplasms, mostly sinonasal inverted papillomas (SNIPs) (*Zhou et al., 2013*; *Lee et al., 2019*; *Zhou et al., 2018*; *Yu et al., 2018*). The benefits of EPLAs over standard endoscopic maxillectomies when treating SNIPs include its ability to preserve as many normal structures as possible (*Turri-Zanoni et al., 2017*), which makes wound healing faster, and its ability to avoid alter turbulent nasal airflow due to preserve of inferior turbinate (*Chen et al., 2010*). It has also been associated with less lacrimal pathway obstruction and fewer thick crusting because more of the epithelium is preserved (*Hildenbrand, Weber & Brehmer, 2011*). In their multicenter retrospective study, *Zhou et al. (2018)* reported recurrence rates in 71 Krous staged III SNIPs patients treated with EPLAs to be similar to those reported by *Lombardi et al. (2011)* performing the largest endoscopically cohort study to date. However, patients in *Zhou et al. (2018)* were spared from post-operative mucocele developments and negative naso-lacrimal complex sequelae (*Lombardi et al., 2011*; *Zhou et al., 2018*). *Bertazzoni et al. (2017)*, using extended endoscopic maxillectomies exclusively, also found significantly greater number for sensory deficits and esthetic problems than *Zhou et al. (2018)* reported. Three studies of EPLA SNIPs resections in relatively small patient groups have recently reported EPLAs to result in favorable local control rates, though they did not investigate post-surgery quality-of-life (*Lee et al., 2019*; *Yu et al., 2018*; *Lin, Lin & Yeh, 2018*). Our eight sinonasal papillomas patients reported a median SNOT-22 score of eight after resection, slightly lower than the scores reported by a previously reported case series (*Harrow & Batra, 2013*). Although statistical significance was not reached in our study, our results suggest improvements quality of life metrics after EPLA in these patients.

The proposed mechanisms for relatively higher incidence of transient neural sequel in our sinonasal papilloma patients are illustrated as the follows. The first is the majority of our study population has type II PLR (AP diameter ranged from 3 mm to 7 mm) (*Simmen et al., 2017*). In an imaging study investigating the feasibility of EPLAs, *Simmen et al. (2017)* concluded more bone removal and probable translation tear sac are needed in type II subjects in comparison to those with type III (AP > 7 mm). More bone removal might imply higher probabilities for nerve insults. Another is our exclusive performance of bone drilling on tumor bed (*Healy et al., 2016*). We hypothesize the direct and indirect nerve injuries due to osculating and heating effects by drillings may be responsible for the relative higher incidence for transient sensory deficit. As for residual inferior meatotomy, *Preti et al. (2019)* proposed an innovative method by performing mucosal incision and osteotomy at different levels to minimize such an adverse event.

Some studies have reported greater/faster symptoms reduction and less disease recurrence following surgeries compared with appropriate pharmaceutical managements alone in certain subgroups of chronic rhinosinusitis (*Fokkens et al., 2019*; *Alsharif et al., 2019*; *Wang, Gullung & Schlosser, 2011*). However, the appropriateness of primary EPLAs to treat lesions due to inflammation in rhinosinusitis has not been addressed. Our reason for extending the use of EPLA to inflammatory maxillary disease is that EPLAs

can offer intuitive viewing and more maneuvering space by zero degree endoscope. Thus, EPLAs may save more time than MMAs without additional complications, especially when treating cases in which inflammatory polyps full of maxillary sinus. In these cases, stripping lesions arising from the anterior and/or medial aspect of maxillary sinus (e.g., alveolar or PLR) present an obstacle for standard MMA approach (*Turri-Zanoni et al., 2017*; *Kashlan & Craig, 2018*). Nonetheless, further studies are needed to elucidate whether patients with inflammatory maxillary disease undergoing EPLAs have results comparable with those treated with conventional MMAs.

This study has some limitations. One limitation is its relatively small sample size and a lack of a control cohort treated with other surgical strategies. This made further comparisons of multiple endpoints among different cohorts with maxillary lesions difficult. Another limitation is that because we only had access to post-operative outcomes approximate 6 months later, we could not determine whether reduced inflammation reached nadir after surgical treatment using EPLA. An additional limitation is that we remain unclear regarding the altered extent of nasal airway resistance and maxillary sinus mucociliary clearance post-operatively. Future investigations are needed to verify long-term rhinologic manifestations. Still another is we did not analyze influencing factors (e.g., laterality *Beswick et al. (2017)*, surgical extent *Ayoub et al. (2019)*, comorbid rhinitis, and concomitant surgical procedures, etc) in the current study. Considering their probable impacts on VAS rating and SNOT-22 scores, future studies are warranted to test their significance in the framework of EPLAs.

## CONCLUSION

In conclusion, we found that in addition to tumors, EPLAs can be used to safely and efficaciously treat various lesions within the maxillary sinus. This approach might be used to replace invasive or time-consuming surgical strategies currently used to manage certain pathologies. We found most common adverse event to be transient cheek/tooth numbness but they usually subsided within 6 months. Thus, with adequate patient counseling, symptoms reduction and improvements in life quality can be expected after receiving surgical treatment of maxillary sinus lesions using EPLAs.

## ABBREVIATIONS

| | |
|---|---|
| **EPLA(s)** | endoscopic prelacrimal recess approache(s) |
| **SNOT-22** | 22-item Sino-Nasal Outcomes Test |
| **VAS** | visual analog scales |
| **MMA** | middle meatal antrostomy |
| **PLR** | prelacrimal recess |
| **AJCC** | American Joint Committee on Cancer |
| **AP** | antero-posterior |
| **NLD** | nasal lacrimal duct |
| **IT-NLD** | inferior turbinate-nasolacrimal duct |
| **SD** | standard deviation |

| IQR | interquartile range |
| SNIP(s) | sinonasal inverted papilloma(s) |

### Funding
The authors received no funding for this work.

### Competing Interests
The authors declare that they have no competing interests.

### Author Contributions
- Yu Hsuan Lin analyzed the data, conceived and designed the experiments, performed the experiments, prepared figures and/or tables, authored or reviewed drafts of the paper, collection and assembly of data, and approved the final draft.
- Wei-Chih Chen analyzed the data, conceived and designed the experiments, performed the experiments, prepared figures and/or tables, authored or reviewed drafts of the paper, collection and assembly of data, and approved the final draft.

### Ethics
The following information was supplied relating to ethical approvals (i.e., approving body and any reference numbers):

The protocol of this study was approved by the institutional review board of Kaohsiung Chang Gung Memorial Hospital (IRB No: 201900695B0).

### Data Availability
The raw measurements are available in the Supplemental Files.

### Supplemental Information
Supplemental information for this article can be found online at http://dx.doi.org/10.7717/peerj.8331#supplemental-information.

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
