# Peer review of "Clinical outcome of endonasal endoscopic prelacrimal approach in managing different maxillary pathologies"

_PeerJ, doi:10.7717/peerj.8331_

## Round 0.1 · original submission · Major Revisions

Your article is interesting and relevant for our journal. However, in the current state it cannot be published due to several issues raised by the reviewers. Please address the reviewer comments on a point by point by basis.

Reviewer 1 ·

Basic reporting

English should be polished by professionals.

Experimental design

Methods, Line 197
“EPLA was performed before the other procedures for patients scheduled to receive endoscopic prelacrimal recess approach for maxillary sinus lesions.” Could you clarify what were “ the other procedures”? Did the surgeons performed middle meatal antrostomy as well as EPLA procedure?

Validity of the findings

Results, line 232 to line 234
According to the manuscript, 21 patients (16 males and 5 females) with 22 lesion sites received surgical procedures utilizing the endonasal endoscopic prelacrimcal recess approach (EPLA). Was there one patient undergoing endoscopic surgery via EPLA bilaterally during the surgery? What was the nature of the bilateral maxillary sinus disease? Or, was there one patient undergoing EPLA for two times? What was the indication for revised endoscopic surgery via EPLA?
Results, line 245 to line 255
According to the manuscript, 11 patient had neoplastic lesions, including 9 papillomas and a organized hematoma, and a cavernous hemangioma. However, on line253, one lacrimal duct adenoid cystic carcinoma and one recurrent buccal cancer were also mentioned. It would be better for a more clarified descriptions of the nature and disease entity of maxillary sinus diseases.

Line262 to line 263
What was the hypothesis to compare the VAS ratings and SNOT-22 scores between patients with and without papillomas? Why didn’t compare the subjective symptoms scores between neoplastic or non-neoplastic group? Why didn’t compare the scores between unilateral and bilateral diseases?

Additional comments

In general, the authors described their own experience in doing EPLA without obvious novel findings. The length of discussion section needs to be shortened.
The percentage of complication is too high compared to reference which should be explained.

·

Basic reporting

This is an interesting study on 21 patients submitted to endoscopic prelacrimal recess approach to treat different pathologies of the maxillary sinus. Post-treatment quality-of-life has been properly analyzed together with general outcomes of surgery.
The manuscript should be revised by an English native speaker in order to improve grammar, spelling (e.g. "infratempora" instead of "infratemporal", line 111) and punctuation.

Experimental design

The study is well conducted. Minor suggestions to improve the manuscript are:
1) The CT-based measurements of the prelacrimal recess (AP diameter and height) should be correlated to the post-operative sequelae encountered, especially those related to sensory nerves injury (transient dental/cheek numbness). A possibile result might be an inversely proportional relationship between the dimension of the prelacrimal recess and the risk to have post-operative neural sequelae. You can retrive information from: Simmen D, et al. Anterior maxillary wall and lacrimal duct relationship - CT analysis for prelacrimal access to the maxillary sinus. Rhinology. 2017 Jun 1;55(2):170-174
2) The surgical technique description may be improved in view of a recently-published technical note where the relation between the mucosal incision and the maxillary osteotomy has been described (Preti A, et al. A Surgical Variant of the Pre-Lacrimal Approach to the Maxillary Sinus. Iran J Otorhinolaryngol. 2019 Sep;31(106):327-328). Following this advice, the risk of post-operative residual inferior meatotomy can be minimized.

Validity of the findings

The paper is well written and the conclusions reasonable. The analysis of post-operative outcomes of endoscopic prelacrimal approach is innovative.

Additional comments

The manuscript is really interesting but should be improved following the suggestions raised in the review process.

Reviewer 3 ·

Basic reporting

Many spelling and grammatical errors strewn throughout the document

Experimental design

From initial reading - it is unclear what the study design is. Only after further reading, can one ascertain that it is a prospective study. It would be good if the authors could clearly define this at the beginning of the article.

Measuring pre-lacrimal complex parameters has no benefit in this study, as the results were not used in a meaningful manner in the article.

Validity of the findings

The study is flawed for the lack of a control group. One might argue that any patient undergoing surgery for the pathologies listed would have improved VAS and SNOT-22 scores, regardless of whether the endoscopic pre-lacrimal approach was used. In fact, if there was a control group, it would very likely show that the control group would have similar VAS and SNOT-22 scores.

It was stated that other surgeries such as surgery for a deviated nasal septum was also performed, if necessary, on patients in the study group. This could counfound the improved VAS and SNOT-22 scores. It would be useful if these patients were identified in the cohort. Furthermore, patients may have had other sinuses, including the maxillary sinus operated on, which may also have affected VAS and SNOT-22 scores. This was not clearly stated in the article.

Follow-up care is vague. There was mention of nasal saline irrigation, but it is unclear if any other medical interventions were utilized. Patients with concurrent allergic rhinitis, for example, may have been started on intranasal corticosteroids which by itself, may lead to improved VAS and SNOT-22 scores

---

## Round 0.2 · accepted · Accept

Your responses to the reviewers were satisfactory. There are a few minor items which can be addressed in the proof.

Reviewer 1 ·

Basic reporting

good

Experimental design

good

Validity of the findings

good

Additional comments

The article is much improved and the authors well answered the question raised by the reviews.
Only minor issues should be corrected as below:

p.11, line 114-115: "1.0-cm slices in the axial view"should be 1.0-mm slices

p.16, line 238-239: “This study found that EPLA-treated patients reported improved VAS ratings SNOT-22 scores over time, with mean improvements of 4.2 and 17.6 points, respectively.” This sentence is unclear, respectively what?

p.17, line 263: for full-filed surveillance should be "full-filled"

·

Basic reporting

The Authors have fully addressed all the issues raised in the revision process and the paper is now suitable for publication.

Experimental design

The study design is correct.

Validity of the findings

The results are reasonable.

Additional comments

The paper has been considerably improved and it is now suitable for publication.